# Particle fluctuations and the failure of simple effective models for many-body localized phases

M. Kiefer-Emmanouilidis[1,2], R. Unanyan[1], M. Fleischhauer[1], J. Sirker[2]

**1** Department of Physics and Research Center OPTIMAS, University of Kaiserslautern, 67663 Kaiserslautern, Germany
**2** Department of Physics and Astronomy and Manitoba Quantum Institute, University of Manitoba, Winnipeg R3T 2N2, Canada
* maxkiefer@physik.uni-kl.de

August 9, 2021

## Abstract

We investigate and compare the particle number fluctuations in the putative many-body localized (MBL) phase of a spinless fermion model with potential disorder and nearest-neighbor interactions with those in the non-interacting case (Anderson localization) and in effective models where only interaction terms diagonal in the Anderson basis are kept. We demonstrate that these types of simple effective models cannot account for the particle number fluctuations observed in the MBL phase of the microscopic model. This implies that assisted and pair hopping terms—generated when transforming the microscopic Hamiltonian into the Anderson basis—cannot be neglected. As a consequence, it appears questionable if the microscopic model possesses an exponential number of exactly conserved *local* charges. If such exactly conserved local charges do not exist, then particles are expected to ultimately delocalize for any finite disorder strength.

# 1 Introduction

It has been conjectured that one-dimensional quantum lattice models with short-range hoppings and interactions enter a many-body localized (MBL) phase for sufficiently strong potential disorder [1–17]. Similar to Anderson localization (AL) in non-interacting systems [18–21], particles in an MBL phase are believed to be localized and particle number fluctuations in any partition of the system in the thermodynamic limit should therefore strictly be bounded. On the other hand, introducing local interactions for the original particles induces exponentially decaying long-range interactions between the Anderson localized eigenstates of the non-interacting model leading to a dephasing. As a consequence, in a quench starting from a product state, Anderson eigenstates which are a distance $\ell$ apart become entangled after a time $t \sim e^{\ell}$. This leads, in particular, to the logarithmic increase of the von-Neumann entanglement entropy of a partition, $S \sim \ell \sim \ln t$, which is considered to be one of the hallmarks of an MBL phase [22, 23].

If the particles in an MBL phase are indeed localized, then this type of physics can be described by effective models [8, 10, 24–26]

$$H_{\text{eff}} = \sum_n \varepsilon_n \eta_n + \sum_{nm} J_{nm} \eta_n \eta_m + \cdots \tag{1}$$

with random energies $\varepsilon_n$ and exponentially many conserved charges $[H, \eta_n] = 0$ with $\eta_n = d_n^\dagger d_n$ being the occupation numbers of the *localized orbitals*. Here $J_{nm}$ are non-local interactions which decay exponentially with the distance between the conserved charges. Due to the assumed localized character of the orbitals, the operators $d_n$ in the effective model are related to the original fermionic operators $c_i$ in the microscopic model by

$$d_n^\dagger = \sum_i \langle n|i \rangle_w \, c_i^\dagger, \tag{2}$$

where $|i\rangle_w$ denotes the Wannier state corresponding to the $i$th lattice site and $|n\rangle$ is a state in the basis of localized orbitals. Note that a representation of a given microscopic Hamiltonian by an effective Hamiltonian as given in Eq. (1) is always possible if no restrictions are placed on the form of the $\eta_n$, in particular, if they are allowed to be non-local [27]. What makes this representation special for the MBL case is that the $\eta_n$ are *all* supposed to be local, i.e., these operators only have support—up to exponentially small tails—on a finite number of adjacent lattice sites. If one wants to take into account the renormalization of the orbitals of a non-interacting Anderson localized system when adding interactions, then one has to ensure that this renormalization does not ultimately lead to delocalized orbitals. Otherwise the statement that the microscopic model can be represented by an effective model of the form (1) becomes meaningless. This is an important point which we will return to later.

In a number of recent publications, we have provided evidence that the spinless fermion model

$$H_{\text{micro}} = -J \sum_j (c_j^\dagger c_{j+1} + h.c.) + V \sum_j n_j n_{j+1} + \sum_j D_j n_j \tag{3}$$

shows particle number fluctuations in a partition which are not bounded in the thermodynamic limit for any finite disorder strength $D$ [28–31]. Here $J$ is the hopping amplitude, $V$ the nearest-neighbor interaction, and the potential disorder is drawn from a box-distribution, $D_j \in [-D/2, D/2]$. Throughout this paper we use $J$ as unit of energy and $J^{-1}$ as unit of time, setting $\hbar = 1$. Furthermore, $n_j = c_j^\dagger c_j$ is the particle number operator. This finding seems to indicate that the microscopic model (3) can never be fully described by an effective model of the type given in Eq. (1) with operators $\eta_n$ which are fully localized. On the other hand, while the model (1) does not show any quasi-particle fluctuations for $\eta_n = d_n^\dagger d_n$ local and conserved, it does show, even in this case, bounded particle fluctuations in the original fermionic basis because the quasi-particles $d_n$ are a local linear combination of the $c_i$ particles, see Eq. (2).

The goal of this study is to understand in detail the differences in the particle number fluctuations between the interacting microscopic model (3), the Anderson case ($V = 0$), and simple effective models of the type shown in Eq. (1). Here we want to already stress that it is not known how to exactly construct the local integrals of motion $\eta_n$—otherwise the MBL problem would be fully solved—and that various approximative schemes have been discussed in the literature [25, 26, 32, 33]. Here we will concentrate on one particular numerical scheme but we will argue that the qualitative findings are generic. Our paper is organized as follows: In Sec. 2 we discuss how we construct the effective model and define the measures used to quantify the particle number fluctuations. In Sec. 3, we present and compare numerical data, obtained by exact diagonalizations, for the time evolution of disorder-averaged fluctuation measures after a quantum quench for all three models. We find, furthermore, that clear qualitative differences between the microscopic model (3) and the effective model (1) emerge if we consider the time-averaged fluctuations in the diagonal ensemble which are an upper bound for the true particle variance. These results are presented in Sec. 4. The final section provides a short summary and a discussion of the remaining open questions.

## 2 Effective models and particle fluctuations in a partition

If a non-ergodic, many-body localized phase of a microscopic Hamiltonian $H$ does exist, then there must be a basis in which this Hamiltonian is diagonal with matrix elements $\langle n|i\rangle_w$ in the transformation (2) which are exponentially decaying away from a localization center. The Hamiltonian in this localized basis then takes the form (1) and has exponentially many local conserved charges. In practice it is, however, a very difficult task to find these conserved charges.

Here we consider a specific approximation to obtain an effective model which takes all interactions between localized orbitals into account but assumes that these orbitals $\eta_n$ are the localized Anderson ($V = 0$) orbitals [32, 34]. I.e., the renormalization of the $\eta_n$ due to interactions is neglected. This approximation is expected to be reasonable, in particular, for small interaction strengths $V$. Furthermore, the results will remain qualitatively valid as long as the renormalized orbitals remain local which is required if the MBL phase is truly localized. For the numerical calculations, the effective model is obtained as follows:

- Construct the many-body Hamiltonian for $V = 0$ for a fixed random disorder configuration.

- Obtain the transformation matrix $U$ which diagonalizes the Hamiltonian for $V = 0$.

- Now, starting from the microscopic interacting t-V model (3), transform it into the Anderson basis using the transformation matrix $U$.

- Keep only the diagonal of this many-body Hamiltonian matrix. These are the contributions which are diagonal in the Anderson basis.

- Use the transformation matrix $U^{-1}$ to transform back into the original basis.

- For the obtained effective model, measures of particle fluctuations in the original microscopic basis can now be calculated and directly compared to the full microscopic model and the Anderson case.

In the effective model constructed in this way, off-diagonal contributions such as assisted hopping terms $\sim \sum_{lmn} \eta_l d_n^\dagger d_m + h.c.$ and pair hopping terms $\sim \sum_{klmn} d_k^\dagger d_l^\dagger d_m d_n + h.c.$, which naturally arise when transforming the interaction part of the microscopic model into the Anderson basis using Eq. (2), are neglected. Put another way, the comparison between the microscopic and the effective model will tell us if it is justified to neglect these terms. Note that all three models are always considered at half filling and for exactly the same disorder configuration. Disorder averages can be obtained by performing these steps many times for different random configurations. Starting from initial product states $|\Psi(0)\rangle$, we calculate time evolutions using these three Hamiltonians and monitor the time dependence of particle fluctuations in a partition of the system. For normalized initial states, the expectation value of an operator $O$ is then given by $\langle O(t)\rangle = \langle \Psi(t)|O|\Psi(t)\rangle$ with $|\Psi(t)\rangle = \exp(-iHt)|\Psi(0)\rangle$. All expectation values shown in this paper are averages over many disorder realizations. Time averages are denoted as $\overline{O}$. We want to stress already here that care has to be taken when exactly the time average is performed, a point which will be important for the following discussions.

In order to investigate particle number fluctuations in these models, we partition the system in two equal halves and calculate the probabilities $p(n,t)$ of having $n$ particles in one partition at time $t$. Based on $p(n)$ we can define the average particle number

$$\langle N \rangle = \sum_n p(n)n \tag{4}$$

and the number variance

$$\Delta N^2 = \langle N^2 \rangle - \langle N \rangle^2 = \sum_n p(n)(n - \langle N \rangle)^2. \tag{5}$$

In addition, we will also consider Rényi number entropies [28]

$$S_N^{(\alpha)} = \frac{\ln \sum_n p^\alpha(n)}{1 - \alpha} \tag{6}$$

where $\alpha \geq 0$ is a real parameter. In the limit $\alpha \to 1$ we obtain, in particular, the von-Neumann number entropy $S_N = -\sum_n p(n) \ln p(n)$ [35–44], and in the limit $\alpha \to 0$ the Hartley number entropy.

# 3 Numerical results for time-dependent fluctuation measures

We use exact diagonalizations (ED) of small systems to investigate the quench dynamics in the full interacting model, the Anderson model, and the effective model—constructed as described in the previous section—starting from a charge density wave state $|\Psi(0)\rangle$ where every second site is occupied.

## 3.1 Entanglement and Number Entropy

We concentrate first on the time evolution of the disorder-averaged entanglement entropy $S$ and number entropy $S_\mathrm{N}$. In Fig. 1, we show results for two different disorder and interaction strengths. In the Anderson case, both the entanglement and the number entropy saturate

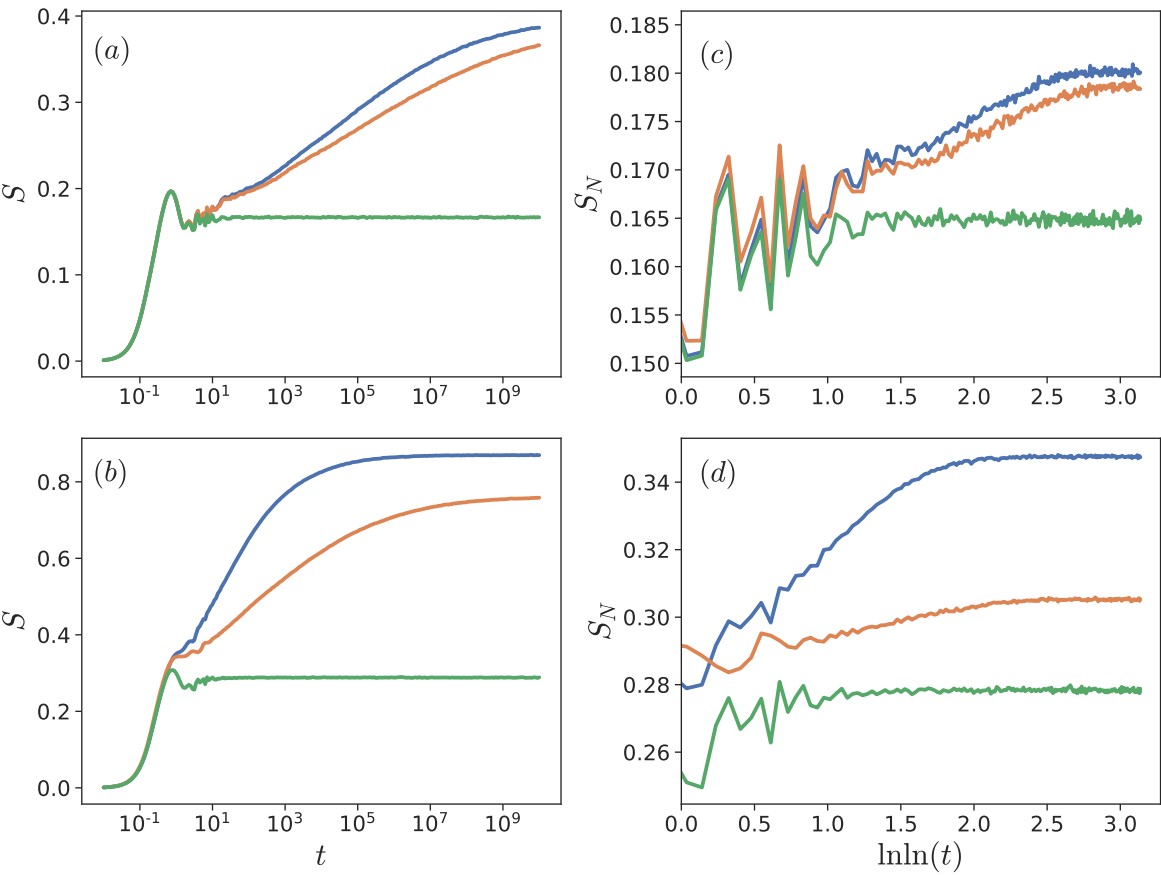

Figure 1: Results for $L = 12$ averaged over 80000 samples with $D = 36$ and $V = 0.2$ (top row) and $D = 20$ and $V = 2.0$ (bottom row). Left column (a, b): entanglement entropy, right column (c, d): number entropy. The largest entropies occur in the full model while the entropies quickly saturate in the Anderson case. The effective model is in between those two cases.

quickly. In the full model, on the other hand, both quantities increase as $S(t) \sim \ln t$ and $S_\mathrm{N}(t) \sim \ln \ln t$ before saturation due to the finite size of the system sets in. For the effective model, we also observe a logarithmic increase of the entanglement entropy for both parameter

sets shown which is expected due to the long-range dephasing terms in Eq. (1). For the number entropy the situation is less clear. While for $D = 36$ and $V = 0.2$, Fig. 1(c), the effective model shows a similar behavior as the full model, this is not the case for $D = 20$ and $V = 2.0$, Fig. 1(d).

Clearly, a more detailed analysis of the scaling of the entropies with system size $L$, disorder strength $D$, and interaction strength $V$ is required. Let us first recapitulate what we have found for the full microscopic model (3): For times $1/V \ll t \ll t_d$ — where $t_d$ is a common deviation time for both $S_N$ and $S_{\text{ent}}$ due to the finite size of the systems studied — we have found that [30, 31]

$$S = \text{const} + \frac{A}{D^3} \ln t, \quad S_N = \text{const} + \frac{B}{D^3} \ln \ln t \qquad (7)$$

with some constants $A, B$. The common deviation time scales as $t_d \sim \exp(L/\xi)/V$ with $\xi \sim 1/\sqrt{D - D_c(V)}$ and is a finite-size effect. For the full microscopic model, the value of $S_N$ where the scaling starts to deviate from $\ln \ln t$ and saturation starts to set in is therefore given by

$$S_N(t_d) = \text{const} + \frac{B}{D^3} \ln\Big(L\sqrt{D - D_c(V)} - \text{const}\Big) \qquad (8)$$

and does depend on $D$, $V$, and $L$.

In the effective model, on the other hand, particle fluctuations only occur inside each localized Anderson orbital, such that $\Delta N^2 \lesssim \xi_A/a$. Here $\xi_A$ is the Anderson localization length and $a$ the lattice parameter. The scaling of the Anderson localization length is known from transfer matrix approaches, $\xi_A = \xi_0/D^2$. Furthermore, we have found [30, 31] that $S_N \sim -\ln\big(1 - 2\Delta N^2\big)$ leading to a saturation value of the number entropy in the effective model given by

$$S_N^{\text{sat}} \sim -\ln\big(1 - 2\xi_0/D^2\big). \qquad (9)$$

Together with the double logarithmic scaling for $S_N < S_N^{\text{sat}}$ this allows to define a saturation time $t_{\text{sat}}$ for the effective model by

$$\text{const} + \frac{B}{D^3} \ln \ln t_{\text{sat}} = -\ln\big(1 - 2\xi_0/D^2\big) + \text{const}. \qquad (10)$$

If $t_{\text{sat}} < t_d$—which will always be true if the system is large enough—then the number entropy in the effective model at long times will saturate to a constant which only depends on $D$ but not on $V$ and $L$. I.e., in sufficiently large systems, we expect a very different scaling behavior in the full and the effective model. For $D = 20$ and $L = 8 - 16$ this difference in scaling can already be observed numerically as shown in Fig. 2. While the saturation value does depend on $V$ and $L$ in the microscopic model, it is independent of $V$ in the effective model and does become independent of $L$ for $L \geq 14$. We conclude that at least this effective model where the conserved charges are simply the unrenormalized Anderson orbitals cannot account for the observed behavior of the number entropy in the microscopic model. However, renormalizing the Anderson orbitals and thus the average localization length could potentially account for the observed $V$ dependence. In this case though, Eq. (10) would still apply for the renormalized and $V$ dependent correlation length $\tilde{\xi}(V)$. For systems large enough such that $t_{\text{sat}} < t_d$ there will then still be an $L$-independent saturation. The only scenario where the effective model could explain the finite-size data is if in the effective model with renormalized—but still local—conserved charges the saturation time $t_{\text{sat}}$ is always larger than the deviation time $t_d$ for all system sizes accessible by ED. Such a scenario can never be ruled out entirely based on numerical data for finite system sizes.

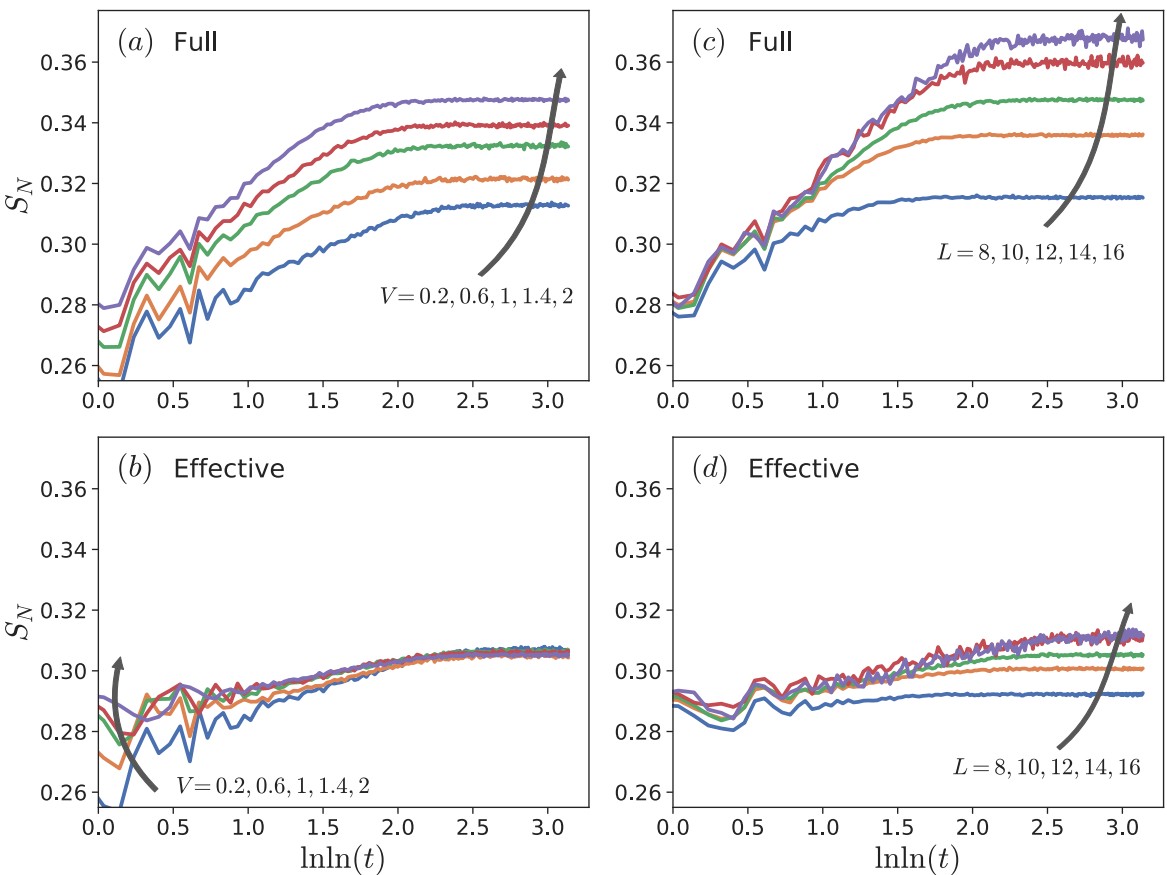

Figure 2: Number entropies for $L = 8, 10$ (200000 samples), $L = 12$ (40000 samples), $L = 14$ (4000 samples), and $L = 16$ (3000 samples) with $D = 20$: The full model is shown in the top row, the effective model in the bottom row. Left column (a, b): dependence on interaction $V$ for $L = 12$, right column (c, d): dependence on length $L$ for $V = 2.0$.

## 3.2 Hartley number entropy

The number entropy is not sensitive to large particle fluctuations occurring with a low probability. As discussed in more detail in earlier publications [30, 31], a better suited quantity is the Hartley number entropy $S_H$, formally obtained from the Rényi number entropy, Eq. (6), in the limit $\alpha \to 0$. However, since the unitary dynamics of the system immediately couples the initial state with all the other states in the same symmetry sector, it is crucial to introduce a cutoff and only include probabilities $p(n) > p_c$ in order to obtain a quantity which measures particle fluctuations in a meaningful way. While the cutoff $p_c$ is arbitrary, the qualitative behavior is the same for different cutoffs as long as they are small. Here we will choose a cutoff $p_c = 10^{-10}$. Furthermore, we cannot take the limit $\alpha \to 0$ exactly in the simulations and instead choose a small fixed parameter $\alpha = 10^{-3}$.

For the Hartley entropy we expect a more pronounced difference between a model where the particle movement is limited to their (renormalized) Anderson orbitals and a model where hopping processes between such orbitals can occur. I.e., in a localized model, $p(n) \sim \exp(-|n - n_{\max}|)$ at long times where $n_{\max}$ is the particle number where the distribution is

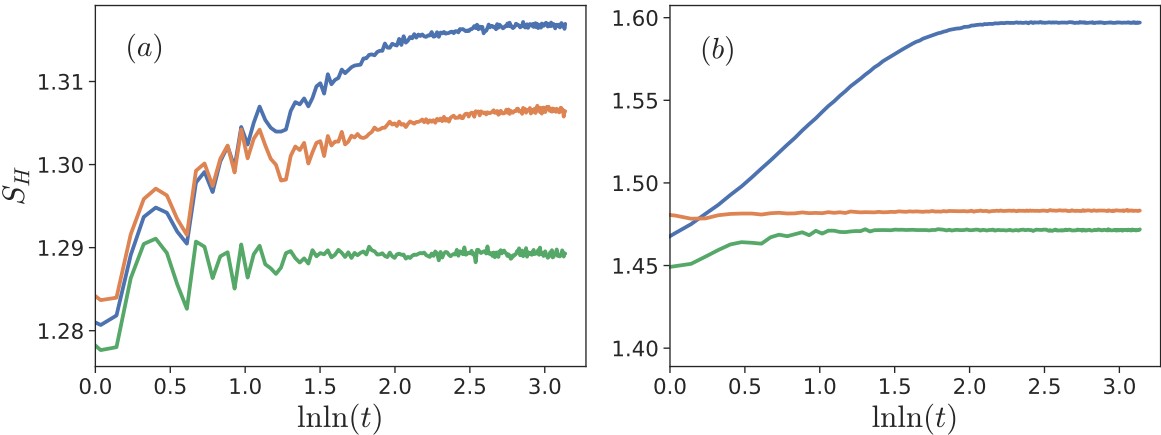

Figure 3: Hartley entropy for $L = 12$ and 80000 samples with $D = 36$ and $V = 0.2$ (left) and $D = 20$ and $V = 2.0$ (right).

maximal. As shown in Fig. 3, we indeed find that $S_H$ saturates quickly in the effective model while the microscopic model shows a $\ln\ln t$ increase up to the deviation time $t_d$. This difference becomes more pronounced with increasing interaction strength $V$.

Furthermore, we find that similar to the number entropy the saturation value of $S_H$ in the effective model is again independent of $V$ and becomes independent of $L$ for $L \geq 12$ while it does depend on both parameters in the full model, see Fig. 4. Note also that in the full model the time scale where the Hartley entropy deviates from the $\ln\ln t$ scaling is again $t_d$ as for the entanglement $S(t)$ and the number entropy $S_N(t)$. I.e., in the microscopic model there is only a single finite-size time scale controlling the dynamics of all entropies.

## 3.3 Particle number fluctuations

Finally, we also want to compare directly the particle number fluctuations $(\Delta N)^2(t)$ in all three models which is the quantity which we will study further in Sec. 4. As for the Rényi entropies, we start by comparing all three models for two different sets of disorder and interactions strengths, see Fig. 5. The results are very similar to those for the number entropy shown in Fig. 1. While the effective model for small system sizes captures the particle number fluctuations well for small interaction strengths $V$ and large disorder $D$, this is not the case for larger system sizes or larger $V$ and smaller $D$.

If we consider again in more detail the scaling with $V$ and $L$ as shown in Fig. 6, then we also find results which are consistent with those for the number entropy shown in Fig. 2. In particular, the particle fluctuations in the effective model at long times are again independent of the interactions strength $V$ in contrast to the full microscopic model. We also observe that $(\Delta N)^2(t \to \infty)$ in the effective model starts to become independent of system size for $L \geq 14$ which is consistent with the results for $S_N$.

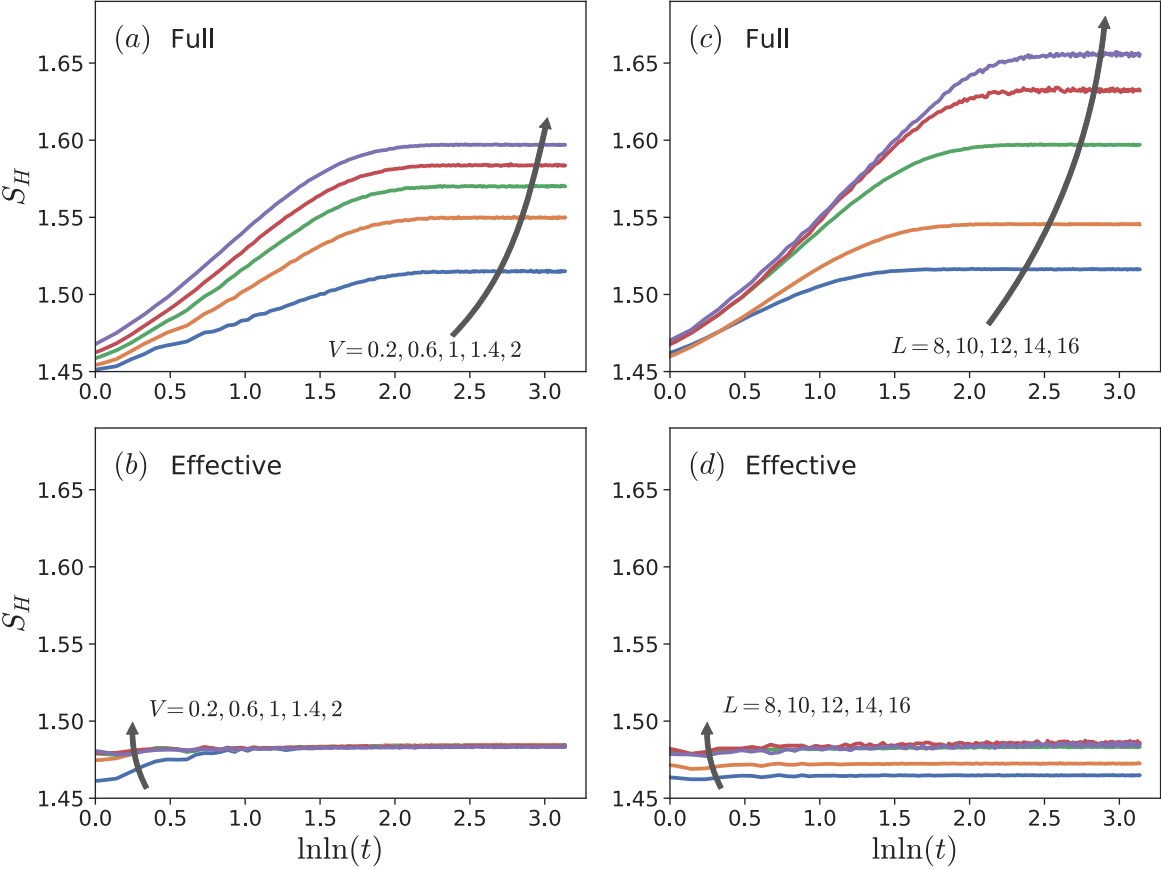

Figure 4: Hartley number entropies for $L = 8, 10$ (200000 samples), $L = 12$ (40000 samples), $L = 14$ (4000 samples), and $L = 16$ (3000 samples) with $D = 20$: The full model is shown in the top row, the effective model in the bottom row. Left column (a,b): dependence on interaction $V$ for $L = 12$, right column (c,d): dependence on length $L$ for $V = 2.0$.

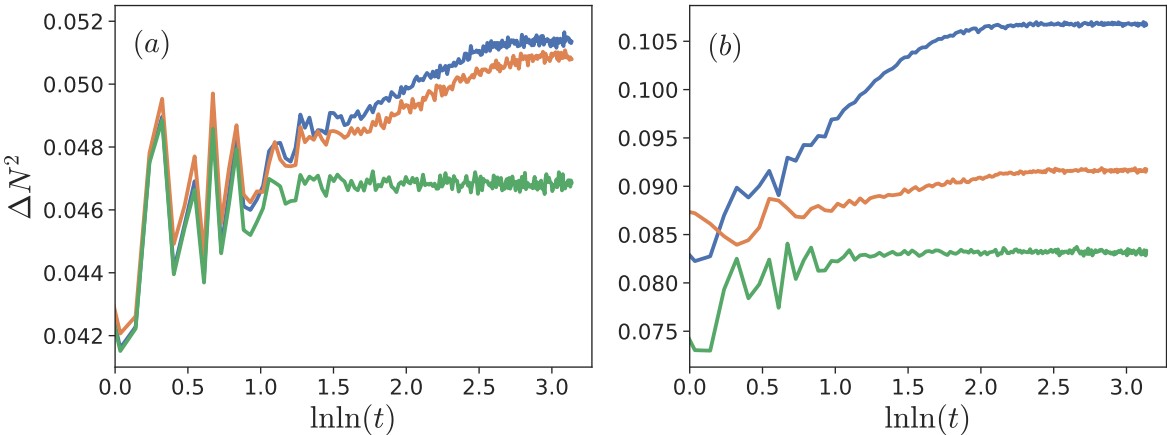

Figure 5: $(\Delta N)^2$ for $L = 12$ and 80000 samples with $D = 36$ and $V = 0.2$ (a) and $D = 20$ and $V = 2.0$ (b).

# 4  Time-averaged number fluctuations

Instead of evaluating the time evolution of disorder-averaged quantities for which it is difficult to attain analytical insights, it is useful to consider time-averaged quantities. The reason why this is helpful is that in the average over infinitely large times only diagonal terms in the eigenbasis survive for linear observables

$$
\begin{aligned}
\overline{\langle O \rangle} &= \lim_{T \to \infty} \frac{1}{T} \int_0^T dt\, \langle \Psi(t)|O|\Psi(t) \rangle \\
&= \sum_{k,m} \langle \Psi(0)|m \rangle \langle m|O|k \rangle \langle k|\Psi(0) \rangle \lim_{T \to \infty} \frac{1}{T} \int_0^T dt\, \exp[i(E_m - E_k)t] \\
&= \sum_k |\langle k|\Psi(0) \rangle|^2\, \langle k|O|k \rangle .
\end{aligned}
\tag{11}
$$

The infinite time average is thus the same as the one obtained when averaging using the diagonal ensemble $\rho_{\mathrm{diag}} = \sum_k p_{\mathrm{diag}}(k)|k\rangle\langle k|$ with $p_{\mathrm{diag}}(k) = |\langle k|\Psi(0)\rangle|^2$. Note that in the last line of Eq. (11) we have assumed that energy eigenvalues are non-degenerate. If this is the case, then only eigenstates enter and the dependence on eigenenergies drops out. Different models with the same diagonal state, such as the Anderson and effective model, then have the same time-averaged expectation values.

## 4.1  Time-averaged characteristic function and diagonal-ensemble number fluctuations

One of the difficulties in analyzing the time-averaged particle fluctuations $\overline{\Delta N^2}$ is that this quantity is not described by a diagonal ensemble. This can be seen as follows:

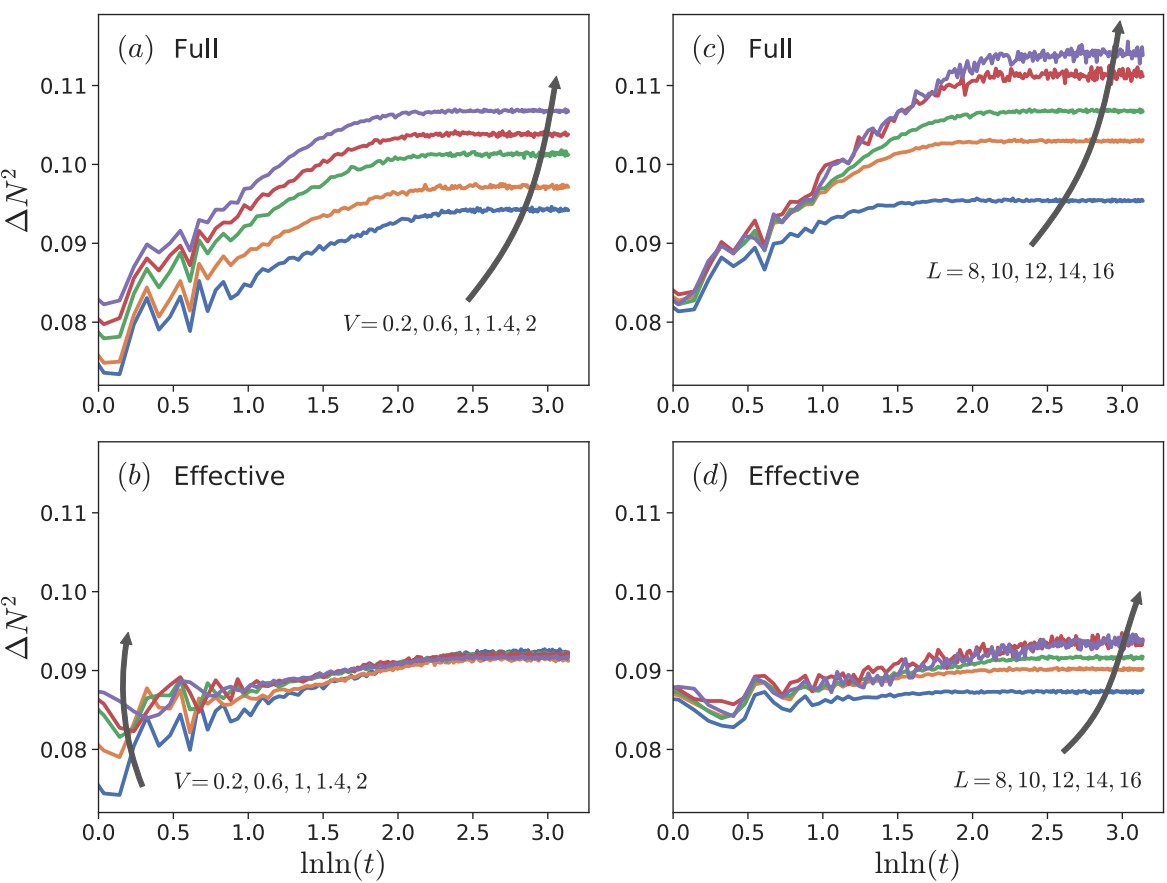

Figure 6: $\Delta N^2(t)$ for full and effective model for different interaction strength $V$ and system sizes $L$. For $L = 8, 10$ 200000 samples were used, for $L = 12$ 40000 samples, for $L = 14$ 4000 samples, and for $L = 16$ 3000 samples, all with $D = 20$. The full model is shown in the top row, the effective model in the bottom row. Left column (a, b): dependence on interaction $V$ for $L = 12$, right column (c, d): dependence on length $L$ for $V = 2.0$.

$$
\begin{aligned}
\overline{\Delta N^2} \quad &= \quad \overline{\langle N^2 \rangle - \langle N \rangle^2} = \overline{\langle N^2 \rangle} - \overline{\langle N \rangle^2} \tag{12}\\[6pt]
&= \quad \lim_{T \to \infty} \frac{1}{T} \int_0^T dt \left\{ \langle \Psi(t)|N^2|\Psi(t)\rangle - \left(\langle \Psi(t)|N|\Psi(t)\rangle\right)^2 \right\} \\[6pt]
&= \quad \sum_{k,m} \langle \Psi(0)|k\rangle \langle k|N^2|m\rangle \langle m|\Psi(0)\rangle \lim_{T \to \infty} \frac{1}{T} \int_0^T dt\, \mathrm{e}^{i(E_k - E_m)t} \\[6pt]
&\quad - \sum_{q,m,k,l} \langle \Psi(0)|q\rangle \langle q|N|m\rangle \langle m|\Psi(0)\rangle \langle \Psi(0)|k\rangle \langle k|N|l\rangle \langle l|\Psi(0)\rangle \\[6pt]
&\quad \times \lim_{T \to \infty} \frac{1}{T} \int_0^T dt\, \mathrm{e}^{i(E_q - E_m + E_k - E_l)t} \;.
\end{aligned}
$$

If we now evaluate the time integrals, then we obtain

$$\overline{\Delta N^2} = \sum_m |\langle \Psi(0)|m\rangle|^2 \langle m|N^2|m\rangle \tag{13}$$

$$- \sum_{\substack{q,m,k,l \\ E_q - E_m + E_k - E_l = 0}} \langle \Psi(0)|q\rangle \langle q|N|m\rangle \langle m|\Psi(0)\rangle \langle \Psi(0)|k\rangle \langle k|N|l\rangle \langle l|\Psi(0)\rangle .$$

If we assume, furthermore, that the system has *no degenerate energy gaps*—a common assumption believed to be true for interacting systems [45–48]—then we can simplify the last term further by using

$$\lim_{T\to\infty} \frac{1}{T} \int_0^T dt\, e^{it(E_q - E_m + E_k - E_l)} = \delta_{qm}\delta_{kl} + \delta_{ql}\delta_{km} - \delta_{qk}\delta_{qm}\delta_{ql} . \tag{14}$$

Note that the condition of non-degenerate energy gaps, i.e. the condition that $E_q - E_m = E_l - E_k$ implies that either $E_q = E_m$ and $E_l = E_k$ or $E_q = E_l$ and $E_m = E_k$, does not restrict the occurance of degeneracies in the energy spectrum itself. While this condition is expected to be true in systems where all subsystems interact with each other, it can be violated in systems where subsystems become independent of the rest of the system. In particular, we expect that this condition is not fulfilled in an Anderson localized system.

If Eq. (14) is fulfilled, then we can write the time average of the particle fluctuations as

$$\overline{\Delta N^2} = \underbrace{\sum_m |\langle \Psi(0)|m\rangle|^2 \langle m|N^2|m\rangle - \left( \sum_m |\langle \Psi(0)|m\rangle|^2 \langle m|N|m\rangle \right)^2}_{=\overline{\Delta \mathcal{N}^2}} \tag{15}$$

$$- \sum_{\substack{k,m \\ k \neq m}} |\langle \Psi(0)|k\rangle|^2 |\langle \Psi(0)|m\rangle|^2 |\langle k|N|m\rangle|^2.$$

We note that even in this case the time averaged particle fluctuations are not described by a diagonal ensemble average which correspond to the first line of Eq. (15) only and which are given by

$$\overline{\Delta \mathcal{N}^2} = \overline{\langle N^2\rangle} - \overline{\langle N\rangle}^2 . \tag{16}$$

The difference of the diagonal-ensemble and time-averaged particle fluctuations is therefore

$$\delta N^2 = \overline{\Delta \mathcal{N}^2} - \overline{\Delta N^2} = \overline{\langle N\rangle^2} - \overline{\langle N\rangle}^2 , \tag{17}$$

i.e., it is due to the order in which the square of the expectation value and the time average are taken. We note that the square function is convex and therefore, due to Jensen's inequality, $\delta N^2 \geq 0$. In spectral representation, this difference corresponds to the last line in Eq. (15) if condition (14) is fulfilled. However, independent of whether or not this condition holds, $\overline{\Delta \mathcal{N}^2}$ is always an upper bound for the true particle fluctuations $\overline{\Delta N^2}$ which is important for the following discussion.

The diagonal-ensemble fluctuations $\overline{\Delta \mathcal{N}^2}$ naturally arise when we consider the time-averaged distribution of particle numbers $N$ in one partition of the system. The number distribution at a time $t$ is fully described by the characteristic function

$$\chi(\theta, t) = \langle \Psi(t)| \exp(-i\theta N)|\Psi(t)\rangle. \tag{18}$$

The number distribution in a time-averaged state is then governed by the *time-averaged* characteristic function

$$\chi_\infty\left(\theta\right) = \lim_{T\to\infty}\left(\frac{1}{T}\int_0^T dt\, \chi\left(\theta,t\right)\right) \tag{19}$$

from which we can obtain all moments, e.g.

$$\overline{\langle N\rangle} = i\frac{\partial}{\partial\theta}\chi_\infty\left(\theta\right)\Big|_{\theta=0} \tag{20}$$

or the variance

$$\overline{\Delta\mathcal{N}^2} = \overline{\langle N^2\rangle} - \left(\overline{\langle N\rangle}\right)^2 = -\frac{\partial^2}{\partial\theta^2}\ln\chi_\infty(\theta)\Big|_{\theta=0}. \tag{21}$$

In the Anderson model, the number fluctuations in one partition of the system after a quench are known to attain a finite asymptotic value independent of system size and so will the number fluctuations $\overline{\Delta\mathcal{N}^2}$ in the corresponding diagonal ensemble. Since the eigenbasis of the considered effective model is identical to the Anderson basis, $\overline{\Delta\mathcal{N}^2}$ in the effective model agrees with that in the Anderson model provided the initial states are the same. As a consequence, the time-averaged number fluctuations $\overline{\Delta N^2}$ in the effective model are bounded from above by a quantity that becomes system-size independent when approaching the thermodynamic limit. We have therefore proven that the particle fluctuations in the effective model are bounded.

Finally, we note that the two fluctuations, $\overline{\Delta N^2}$ and $\overline{\Delta\mathcal{N}^2}$ agree in the thermodynamic limit if the condition of non-degenerate energy gaps (14) is fulfilled and if $|\langle\Psi(0)|m\rangle|^2 \sim 1/\Omega$ where $\Omega$ is the dimension of the Hilbert space. In this case, the second line in Eq. (15) will go to zero. We can expect the latter condition to be fulfilled for typical initial states $|\Psi(0)\rangle$ which have an overlap with a macroscopic number of eigenstates $|m\rangle$. This point is discussed further in the Appendix.

## 4.2   Numerical results

According to the results derived above, the time-averaged particle fluctuations in the diagonal ensemble $\overline{\Delta\mathcal{N}^2}$ are identical in the effective model and in the Anderson model. I.e., the fluctuations do not change when adding interactions as long as the Anderson orbitals remain unchanged and the interaction is diagonal in those orbitals. This is confirmed by the exact diagonalization results shown in Fig. 7 where we perform in addition an average over all initial states which we indicate by $\langle\langle\cdots\rangle\rangle_i$.

The long-time average is increasing with system size for the full microscopic model while it is decreasing towards a finite asymptotic value in the thermodynamic limit for the Anderson and the effective model. This decrease of the long-time average in the Anderson and the effective model is a consequence of a $1/L$ correction when averaging over all initial product states. It is caused by certain initial states where, for example, all particles are initially in one half of the system, see the Appendix for a more in depth discussion. These $1/L$ corrections also affect the scaling of the number fluctuations in the full microscopic model and make it harder to analyze the finite-size scaling. In fact, these corrections might be the reason that the increase of the number fluctuations with system size in the interacting model has been overlooked in the past. If an average over all initial states is performed, then the finite-size scaling in the interacting model should better be considered relative to those in the non-interacting case. This largely eliminates the common $1/L$ corrections and shows that the relative fluctuations

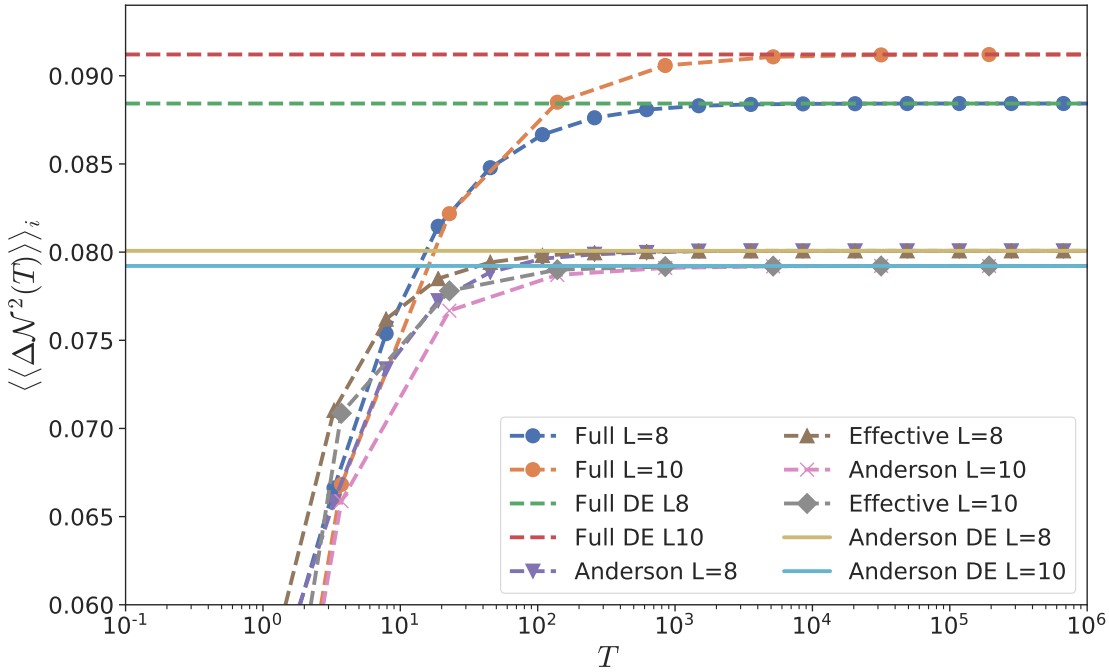

Figure 7: Particle fluctuations in a partition averaged over all intial states $\langle\langle\Delta\mathcal{N}^2(T)\rangle\rangle_i = \frac{1}{T}\int_0^T dt\,\langle\langle\Delta\mathcal{N}^2(t)\rangle\rangle_i$ for the interacting spinless fermion model with disorder compared to those in the Anderson and the effective model. 10000 disorder realizations are used for $L = 8$, and 5000 for $L = 10$.

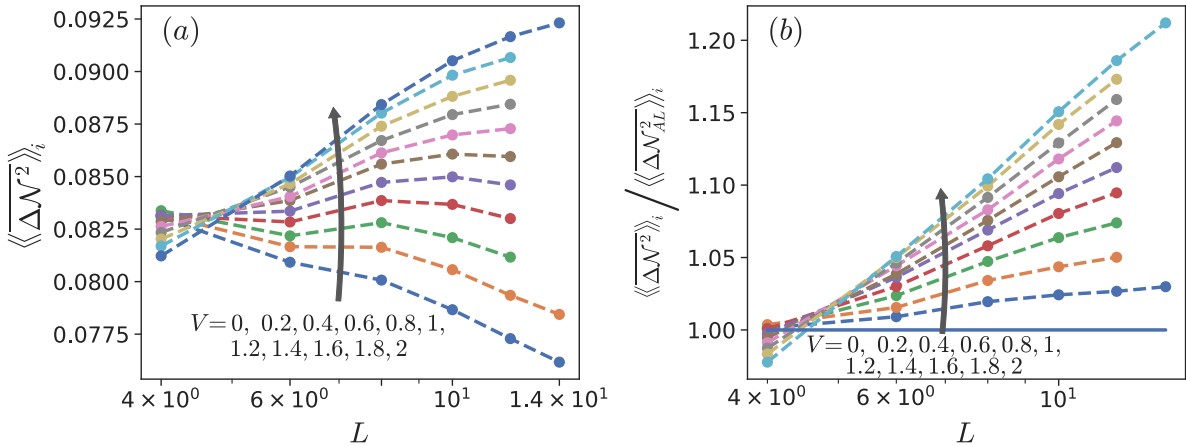

Figure 8: (a) Time-averaged particle fluctuations $\langle\langle\overline{\Delta\mathcal{N}^2}\rangle\rangle_i$ in dependence of system size $L$ averaged over all possible initial states. (b) Same as in (a) but relative to the fluctuations in the non-interacting case $\langle\langle\overline{\Delta\mathcal{N}_{AL}^2}\rangle\rangle_i$. The data are averaged over 10000 disorder realizations.

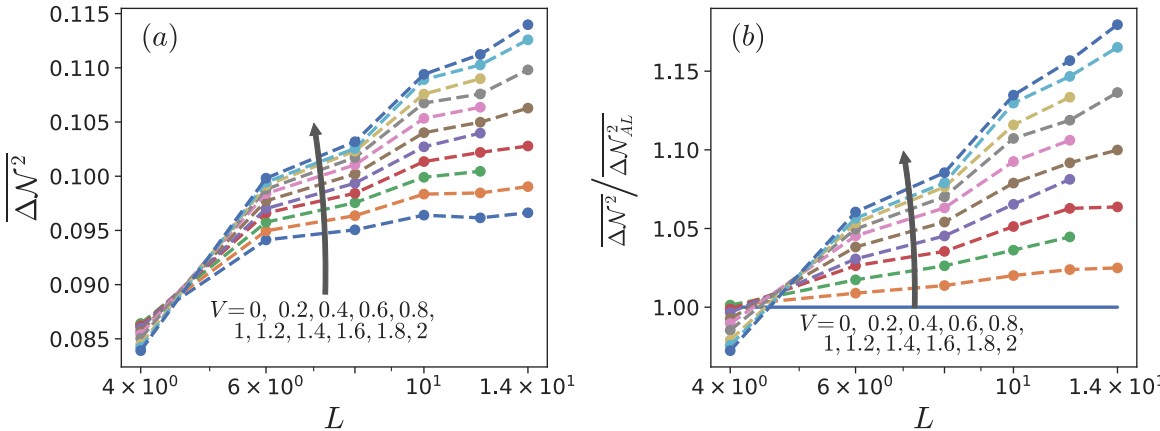

Figure 9: Same as Fig. 8 with the charge-density wave state as initial state instead of averaging over all initial product states. For $L \leq 10$ the data is averaged over 50000 disorder realizations, 20000 for $L = 12$, and 10000 for $L = 14$.

increase roughly with a power law or logarithmically with $L$, see Fig. 8. Alternatively, we can pick a typical initial product state such as the charge density wave state studied earlier where every second site is occupied. In this case, the finite-size corrections in the non-interacting case are much smaller, see Fig. 9.

The numerical data presented here clearly demonstrate that the particle fluctuations in the microscopic model—even for the small system sizes accessible in ED—cannot be accounted for by the effective model which has particle fluctuations which are finite in the thermodynamic limit and identical to those in the Anderson model.

Let us now return to the relation between the true particle fluctuations $\overline{\Delta N^2}$ and those in the diagonal ensemble $\overline{\Delta \mathcal{N}^2}$. We start by numerically investigating the validity of the assumption (14) of non-degenerate energy gaps. In Fig.10(a), a comparison between $\Delta N^2$ calculated with and without this assumption is shown for all three models. We note first that $\Delta N^2$ obtained from Eq. (15), i.e. assuming that Eq. (14) is valid, is identical for the Anderson and the effective model because this quantity only depends on the eigenstates which remain unchanged. However, while this quantity appears to become identical to the time-averaged number fluctuations $\overline{\Delta N^2}$ in the thermodynamic limit for the effective model, this is not the case for the Anderson model. Physically, this can be understood as follows: While the Anderson model separates into subsystems which—up to exponentially small contributions— are independent, the interaction present in the effective model does couple these subsystems. The assumption of non-degenerate energy gaps, Eq. (14), therefore fails for the Anderson model while it appears to be fulfilled for the effective model due to the interaction induced dephasing. For the full microscopic model the condition (14) also appears to hold.

In addition, we also expect that for typical initial states the last line in Eq. (15) goes to zero in the thermodynamic limit. I.e., for the effective and the full microscopic model we expect that $\Delta \mathcal{N}^2 \to \Delta N^2$ for $L \to \infty$. Fig. 10(b) confirms this expectation showing that the difference between the two fluctuation measures goes to zero exponentially with system size.

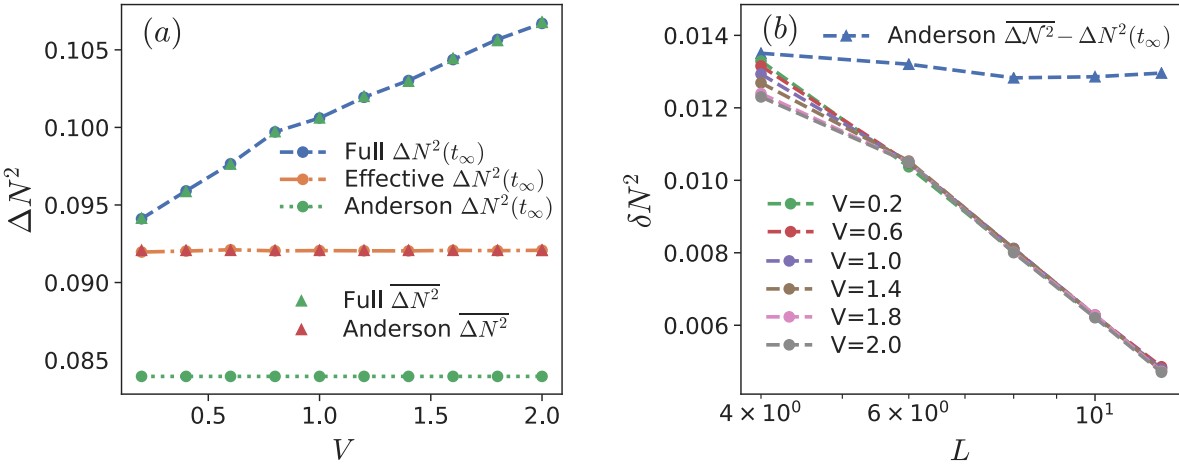

Figure 10: (a) Comparison of the time-averaged fluctuations $\overline{\Delta N^2}$ obtained from Eq. (15) where the assumption (14) has been used (triangles) with $\Delta N^2(t \to \infty)$ (dots) where the assumption (14) has not been used for the full microscopic model, the effective model, and the Anderson model for $L = 12$. (b) $\delta N^2$, Eq. (17), for the microscopic model and different interaction strengths. In all cases, the data are consistent with $\delta N^2 \sim \exp(-L)$. For comparison, the Anderson case is shown as well. For $L \leq 10$ the data is averaged over 50000 disorder realizations, and 20000 for $L = 12$.

## 5  Summary and Conclusions

In this paper, we have investigated the particle number fluctuations in a partition of the t-V model with potential disorder. We have compared the results with the non-interacting Anderson case, and with an effective model with exponentially many local charges, obtained by only keeping interaction terms which are diagonal in the Anderson basis. Using various measures for the particle fluctuations such as the number and Hartley entropies as well as the variance, we have found that there are quantitative and qualitative differences when comparing the time evolution after a quantum quench for the interacting microscopic model and the effective model. In particular, while the number fluctuations in the microscopic model depend on interaction strength and increase as a function of system size for a fixed disorder strength, they are independent of interaction strength at long times for the effective model and become independent of system size, i.e. they saturate to a finite value, for the largest system sizes considered.

   To investigate the difference in the particle number fluctuations between these two models further, we have shown that the time-averaged particle number variance $\Delta N^2 = \overline{\langle N^2 \rangle} - \overline{\langle N \rangle^2}$ can be bounded from above by the variance $\overline{\Delta \mathcal{N}^2} = \overline{\langle N^2 \rangle} - \overline{\langle N \rangle}^2$ obtained from the time-averaged characteristic function. The latter is entirely determined by a diagonal-ensemble average, while the first is not. I.e., $\overline{\Delta \mathcal{N}^2}$ is independent of the eigenenergies and diagonal in the eigenstates. Furthermore, we have shown that the difference between the two fluctuation measures, $\delta N^2 = \overline{\Delta \mathcal{N}^2} - \overline{\Delta N^2}$, vanishes in the thermodynamic limit if the system does not have degenerate energy gaps and if we start the quench from a typical state which does have non-vanishing overlaps with a macroscopic number of eigenstates. For the fluctuation measure $\overline{\Delta \mathcal{N}^2}$ a clear, qualitative difference between the microscopic and the effective model

then emerges: while the fluctuations in the effective model are exactly the same as in the Anderson model, do not depend on interaction strength, and do not increase with system size, the fluctuations in the microscopic model are larger and do increase with system size.

Thus, clearly, the studied effective model does not account for the observed increase of the particle number fluctuations with system size in the microscopic model. In other words, when expressing the microscopic model in the Anderson basis using the transformation (2), off-diagonal terms describing assisted and pair-hopping processes—which naturally arise from the interaction—cannot be neglected. The question then is, whether these terms simply renormalize the Anderson orbitals while still allowing for an effective description of the form (1) with *local* conserved charges or whether such a renormalization ultimately leads to these charges becoming non-local. In the latter case, the disordered many-body system would not be localized. Based on the numerical data for the accessible system sizes we believe it is fair to say that there is no evidence that $\Delta \mathcal{N}^2$ and therefore $\Delta N^2$ in the disordered t-V model is bounded. We note, furthermore, that performing averages over all initial product states leads to a $1/L$ correction to $\Delta \mathcal{N}^2$ with a negative sign which is present already in the non-interacting Anderson case and which can disguise the increase of $\Delta \mathcal{N}^2$ with system size in the interacting case. This might explain why this increase has been missed in the past. Lastly, we remark that if we assume that the overlap of a typical initial state with each eigenstate is $\sim 1/\Omega$, where $\Omega$ is the dimension of the Hilbert space, then $\Delta \mathcal{N}^2 \sim \frac{1}{\Omega} \sum_m \langle m|N^2|m\rangle - \frac{1}{\Omega^2} (\sum_m \langle m|N|m\rangle)^2$. I.e, under this assumption the question whether or not $\Delta \mathcal{N}^2$ is bounded is reduced to an investigation of the fluctuations in the eigenstates of the system which might be a useful simplification for further investigations.

## Acknowledgement

We would like to thank Guiseppe De Tomasi for fruitful and stimulating discussions. M.K-E., R.U., and M.F. acknowledge financial support from the Deutsche Forschungsgemeinschaft (DFG) via SFB TR 185, Project No.277625399. J.S. acknowledges support by the National Science and Engineering Council (NSERC, Canada) and by the DFG via Research Unit FOR 2316. The numerical simulations were executed on the GPU nodes of the high performance cluster "Elwetritsch" at the University of Kaiserslautern which is part of the "Alliance of High Performance Computing Rheinland-Pfalz" (AHRP) and on Compute Canada high-performance clusters. We kindly acknowledge the support of the RHRK and of Compute Canada.

## Appendix

### A.1    Relation between $\overline{\Delta N^2}$ and $\overline{\Delta \mathcal{N}^2}$

The number fluctuation $\overline{\Delta \mathcal{N}^2} = -\partial_\theta^2 \ln \chi_\infty(\theta)\big|_{\theta=0}$ obtained from the time-averaged characteristic function are identical to the number fluctuations in the diagonal ensemble $\rho_{\mathrm{diag}} = \sum_m p_{\mathrm{diag}}(m)|m\rangle\langle m|$ with probabilities $p_{\mathrm{diag}}(m) = |\langle m|\Psi(0)\rangle|^2$ determined by the initial state $|\Psi(0)\rangle$. They are an an upper bound to the time-averaged number fluctuations $\overline{\Delta N^2}$, since

$$\delta N^2 = \overline{\Delta \mathcal{N}^2} - \overline{\Delta N^2} = \overline{\langle N\rangle^2} - \overline{\langle N\rangle}^2 = \overline{\left(\langle N\rangle - \overline{\langle N\rangle}\right)^2} \geq 0. \tag{22}$$

In the following we show that $\delta N^2$ vanishes in the thermodynamic limit if condition (14) holds. To this end, we note that in this case $\delta N^2$ can be written as (see Eq. (15))

$$\delta N^2 = \sum_{\substack{k,m \\ k \neq m}} |\langle \Psi(0)|k \rangle|^2 |\langle \Psi(0)|m \rangle|^2 |\langle k|N|m \rangle|^2 = \sum_{k,m} p_{\text{diag}}(k) p_{\text{diag}}(m) C_{km} \tag{23}$$

where $C$ is the non-negative, symmetric matrix of overlaps

$$C = \begin{bmatrix} 0 & |\langle 1| N |2 \rangle|^2 & . & . & |\langle 1| N |\Omega \rangle|^2 \\ |\langle 1| N |2 \rangle|^2 & 0 & . & . & |\langle 2| N |\Omega \rangle|^2 \\ . & . & & & . \\ . & . & & & . \\ |\langle 1| N |\Omega \rangle|^2 & |\langle 2| N |\Omega \rangle|^2 & . & . & 0 \end{bmatrix}. \tag{24}$$

$\Omega$ is the dimension of the restricted Hilbert space with fixed total number of particles. In our case $\Omega = L! / \left[\left(\frac{L}{2}\right)!\right]^2$, which in the thermodynamic limit $L \to \infty$ grows exponentially $\Omega \sim L^{-1/2} 2^L$.

The maximum eigenvalue of $C$ is finite and can be bounded by [49]

$$\min_m \left(\Delta N_m^2\right) = \min_m \sum_k C_{mk} \leq \lambda_{\max}(C) \leq \max_m \sum_k C_{mk} = \max_m \left(\Delta N_m^2\right), \tag{25}$$

where

$$\Delta N_m^2 = \langle m| N^2 |m \rangle - \langle m| N |m \rangle^2 \leq \gamma L^2, \tag{26}$$

is the fluctuation of particle number in the chosen partition in the eigenstate $|m\rangle$. $\gamma$ is a system-size independent constant. Hence $\delta N^2$ can be bounded from above by

$$\delta N^2 \leq \lambda_{\max}(C) \sum_m p_{\text{diag}}(m)^2. \tag{27}$$

$\sum_m p_{\text{diag}}(m)^2$ is the inverse participation ratio. In general, a typical initial state $|\Psi(0)\rangle$ overlaps with many eigenstates of the system making $\sum_m p(m)^2$ very small. Indeed, as has been shown in [50], when averaging over all initial states with fixed total number of particles $N_0 = L/2$, denoted by $\langle\!\langle \dots \rangle\!\rangle_i$ one finds

$$\left\langle\!\!\left\langle \sum_m p_{\text{diag}}(m)^2 \right\rangle\!\!\right\rangle_i < \frac{2}{\Omega}. \tag{28}$$

Thus

$$\delta N^2 < \frac{2}{\Omega} \max_m \left(\Delta N_m^2\right) \xrightarrow[L \to \infty]{} 0. \tag{29}$$

We conclude that—provided the condition of non-degenerate energy gaps (14) is fulfilled—the time-averaged particle number fluctuations averaged over all initial product states, $\langle\!\langle \overline{\Delta N^2} \rangle\!\rangle_i$, and the corresponding diagonal-ensemble fluctuations $\langle\!\langle \overline{\Delta \mathcal{N}^2} \rangle\!\rangle_i$ approach each other exponentially with increasing system size $L$. Since condition (14) is fulfilled for the effective and the full microscopic model we found indeed $\delta N^2 \sim e^{-L}$ as shown in Fig.10(b), while $\delta N^2 > 0$ remains finite in the Anderson model.

## A.2    Influence of initial states on scaling of number fluctuations

In Fig. 7 we have seen that the diagonal-ensemble number fluctuations when averaged over all initial product states decrease when going from $L = 8$ to $L = 10$ for the Anderson and effective model. This scaling behavior, which points to a potential problem when analyzing data obtained after averaging over initial states, is at first glance surprising and different from the interacting model. It is an artifact of initial states with large number fluctuations.

In Fig. 11 we have plotted the diagonal-ensemble fluctuations for the Anderson model as function of system size $L$ for an initial density-wave state, i.e. a state where every second site is occupied, and for the case of an average over all random initial states. While for the initial density-wave state $\overline{\Delta\mathcal{N}^2}$ increases with system size towards an asymptotic value which is quickly reached, as naively expected, it *decreases* towards a system-size independent value when we average over all initial states. This $\sim 1/L$ decrease is due to rare initial states with

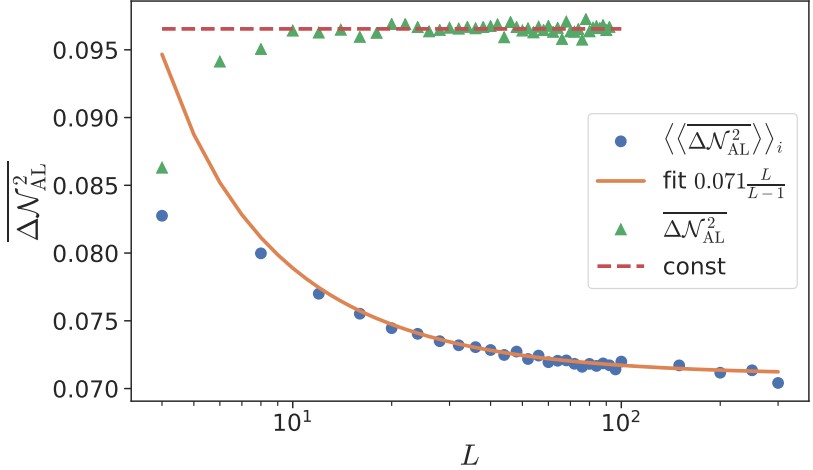

Figure 11: Particle fluctuations in a partition for the Anderson model for an initial density-wave state $\overline{\Delta\mathcal{N}^2}$ (green triangles) and when averaged over all initial states $\langle\!\langle\overline{\Delta\mathcal{N}^2}\rangle\!\rangle_i$ (blue dots). To perform the initial state average we used Eq. (31). The data has been averaged over 50000 disorder realization for $L \leq 100$ and 10000 for $L > 100$.

large number fluctuations whose relative weight becomes smaller with increasing system size. This is illustrated in Fig. 12 where we show a histogram of particle fluctuations in eigenstates of the Anderson model. One clearly recognizes that eigenstates with large fluctuations of the particle number have a much larger probability in smaller systems. When an average over all initial states is taken, also states are included that have a sizable overlap with these eigenstates which results in a larger value of $\langle\!\langle\overline{\Delta\mathcal{N}^2}\rangle\!\rangle_i$ for small systems.

The scaling $\langle\!\langle\overline{\Delta\mathcal{N}^2}\rangle\!\rangle_i \sim L/(L-1)$ seen in Fig. 11, can be understood from a toy model of localization. Let us consider a system with adjacent, spatially non-overlapping, localized orbitals extending over exactly two lattice sites. We cut the system in two partitions assuming that the cut splits the central orbital into two halves and calculate the fluctuations of particle numbers in one partition in an arbitrary eigenstate. The eigenstates are product states of all orbitals occupied by zero, one or two particles with the constraint of a fixed total particle number. We here assume half filling, i.e. $N = L/2$ particles in $L$ lattice sites. Then only

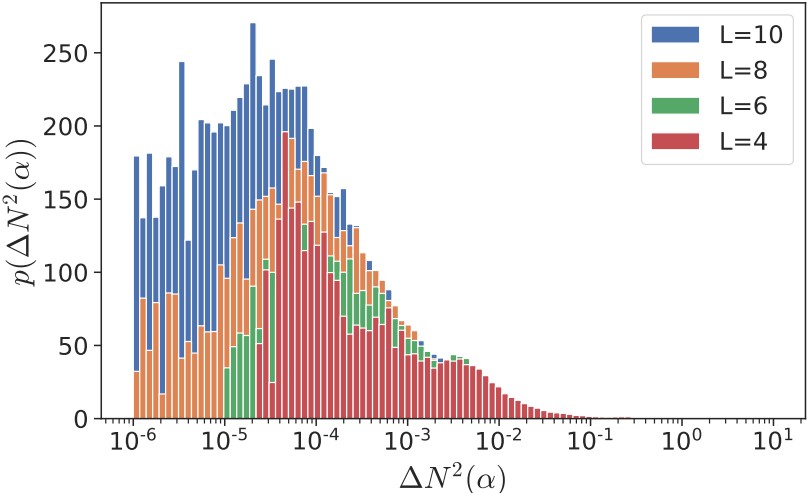

Figure 12: Histogram of particle number fluctuations in eigenstates $|\alpha\rangle$ of the Anderson model for different system sizes $L$. One clearly recognizes that eigenstates with large fluctuations become less and less important when increasing the system size. In all plots we have used 5000 disorder realizations.

those eigenstates contribute to the number fluctuations where exactly one particle is in the central orbital. The probability of such states can easily be computed from combinatorics. It is given by the number of eigenstates where $L/2 - 1$ particles are distributed among the $L - 2$ remaining lattice sites outside of the central orbital divided by the total number of states, leading to

$$\langle\langle\overline{\Delta\mathcal{N}^2}\rangle\rangle_i \sim \frac{\binom{2}{1}\binom{L-2}{L/2-1}}{\binom{L}{L/2}} = \frac{L}{2(L-1)}. \tag{30}$$

A more rigorous derivation of the particle number fluctuations $\langle\langle\overline{\Delta\mathcal{N}^2}\rangle\rangle_i$ averaged over initial states can be done for the Anderson model [51]. This leads to the following expression

$$\langle\langle\overline{\Delta\mathcal{N}^2}\rangle\rangle_i = \frac{L^2}{8(L-1)}\left(1 - \frac{2}{L}\sum_{m=1}^{L}\langle\langle\alpha_m^2\rangle\rangle\right), \tag{31}$$

where $\langle\langle\dots\rangle\rangle$ is used to stress that a disorder average is taken over $\alpha_m^2$ with

$$\alpha_m = \sum_{p=1}^{L}\left|\langle p|m\rangle_w\right|^2 W_{pp}, \tag{32}$$

where $W_{pp} = \langle p|\left[\sum_{j=1}^{L/2}|j\rangle_w {}_w\langle j|\right]|p\rangle$ is the overlap of the $p$th Anderson orbital with the partition of length $L/2$. Eq.(31) is also used in the numerical simulation of the diagonal-ensemble number fluctuations averaged over all initial states, shown in Fig. 11. In the thermodynamic limit, the $\langle\langle\alpha_m\rangle\rangle$ can be approximated as

$$\langle\langle\alpha_m\rangle\rangle \approx \begin{cases} 1 - \beta_m, & \text{for} \quad m \le \frac{L}{2} \\ \beta_m, & \text{for} \quad m > \frac{L}{2} \end{cases}, \tag{33}$$

where

$$\beta_m = \sum_{p=1}^{L} \frac{C_p}{\exp\left[1/4l(p)\right] - 1} \exp\left\{ -\frac{|m - L/2|}{4l(p)} \right\}. \tag{34}$$

$C_p$ is a normalization constant of order unity and $l(p)$ is the localisation length of the $p$th Anderson orbital

$$\left| \langle p|k \rangle_w \right|^2 \sim \exp\left( -\frac{|p - k|}{4\, l(p)} \right), \qquad \text{for} \quad |p - k| \gg l(p). \tag{35}$$

From Eq. (31) one can then derive the following upper and lower bounds

$$\frac{L}{L-1} \frac{l_{\min}}{2} \exp\left( -\frac{1}{4l_{\min}} \right) \leq \langle\!\langle \overline{\Delta \mathcal{N}^2} \rangle\!\rangle_i \leq \frac{L}{L-1} l_{\max}, \tag{36}$$

where $l_{\min}(l_{\max})$ is the minimum (maximum) of $l(p)$.

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
