# Peer review of "Particle fluctuations and the failure of simple effective models for many-body localized phases"

_SciPost Physics_

## Round 1 · Referee Report · Anonymous (Referee 2) · 2021-9-10

Strengths

1- This work probes in a more thorough way an approximation scheme previously introduced in the literature for conserved quantities in interacting one-dimensional systems in a disordered potential.

2- The work looks at interesting quantities to characterise the dynamical behaviour of the system.

Weaknesses

1- Some statements are formulated in a confusing way, or need to be better motivated.

2- Some context and account of previous literature is missing.

Report

This paper deals with the numerical study of particle number fluctuations in finite-size fermionic systems with disorder. The Authors consider three different models:

(i) a canonical model for Many-Body Localization (MBL), with pairwise interacting fermions in a random potential;
(ii) the non interacting, Anderson version of the model;
(iii) an approximate version of (i), which is expected to be provide good results in the weakly-interacting regime.

They compare the dynamical and the time-averaged behaviour of the particle number fluctuations, measured via the von-Neumann and Hartley number entropy, and the number variance. They argue that the approximate model (iii), while reproducing fairly accurately the dynamics of the half-system entanglement entropy of model (i), it does not account for the behaviour of its particle number fluctuations. Based on this observation, they comment on the emergent integrability of the model (i) and on its localization properties. This manuscript belongs to a series of numerical works by the same Authors, Refs. [28-31], in which the occurrence of a truly MBL phase in the thermodynamic limit is considered critically.

I believe that this work is potentially interesting, as it probes certain approximate schemes previously introduced [32,34] by looking at particle number fluctuations, which are interesting quantities to characterize the dynamics of interacting systems in finite-size. Nevertheless, I find that the wording and the presentation of the numerical results are sometimes confusing: I would therefore strongly suggest to the Authors to better motivate certain claims (or eventually to soften them) before publication, in order to “Provide citations to relevant literature in a way that is as representative and complete as possible” and to “summarize the results with objective statements on their reach and limitations”, as required by the SciPost criteria. More detailed comments are given below.

Comments on the presentation of the results:

1) In the abstract and throughout the paper, the Authors insist on the fact that MBL systems are characterised by the presence of an exponential number of local charges. I believe that this statement is confusing (I interpret it as “exponentially-many in the system size L”). Integrability in MBL is usually understood in terms of linearly-many local conserved quantities, which are obtained as quasi-local rotations of the L conserved quantities associated to the quadratic, non-interacting part of the Hamiltonian. Could the Authors specify what they mean by “exponential number” here?

2) In Sec. 2 the Authors define the different Hamiltonians (i-iii) and write that they are used to determine the time evolution. I would clarify here how this time evolution is determined: is it via exact diagonalisation of the Hamiltonian and explicit calculation of the evolution operator? Given that the Authors consider quenches from a fixed pure initial state, could for example Krylov methods be used to have access to larger system sizes (L=18)?

3) In Figs. 4 (d) and 5 (d) the Authors show the scaling with system size of the Hartley number entropy and of the number variance for model (iii), and claim that the curves are overlapping for the two larger system sizes. I suggest to modify (diminish) the y-range of these plots, to make this sister-size independence clearly evident from the plot. The same for Figs. 4 (c) and 5 (c).

4) In Sec. 3.1 the Authors claim a different L dependence of the saturation values of the number entropy in models (i) and (iii): is it correct to say that these conclusions rely on the double-logarithmic fit of the number entropy behaviour in time? Is there any analytical argument justifying this scaling, similar to what one has for the log(t) behaviour of the entanglement entropy?

Comments on the scientific context and interpretation of the results:

5) The approximate model (iii) considered in this work is obtained neglecting the assisted and pair hopping-terms that appear in the Hamiltonian of model (i), when the latter is expressed in the Anderson basis. The numerical data presented in this work indicate that these terms play a relevant role when considering particle number fluctuations and therefore can not be neglected even when the interaction term is not too strong, at least in these finite sizes. Based on this, the Authors also claim: “as a consequence, it appears questionable if the microscopic model possessed an exponential number of exactly conserved local charges”. This implication is not clear to me. At the level of the local conserved quantities , the approximation scheme discussed in this work corresponds to projecting the conserved operators onto a small subspace, that is the kernel of the commutator . This is a quite substantial approximation to the “true” conserved quantities: its failure in reproducing certain features of the dynamics is in my view not sufficient to question overall the existence of local (non-approximated) conserved operators, and thus to question MBL integrability. Could the Authors make their argument more stringent?

6) In the Introduction, it is written “if one wants to take into account the normalisation of the orbitals [….], then one has to ensure that this renormalisation does not ultimately lead to delocalised orbitals”. Similarly, in the Conclusion the Authors write “The question then is, whether […] such a normalization ultimately leads to charges becoming non-local”. In fact, this question has been the subject of several previous works, cited in the introduction of the manuscript. In particular, Ref. [11] claimed a proof of the fact that the renormalisation procedure does not ultimately lead to delocalized orbitals, at least for a particular class of locally-interacting 1d spin models. The same is claimed based on approximate perturbative arguments for fermonic Hamiltonians of the type discussed in this manuscript. I believe that this previous work should be accounted for when raising this point, even more so if the Authors have reasons to challenge those results or interesting intuitions on possible loopholes in the proof or in the perturbative arguments: at the moment, the way the discussion is formulated gives the impression that this issue has remained completely unadressed in the literature.

7) I would also suggest to be more explicit one the fact that this manuscript is dealing with a specific approximation, the one leading to model (iii), and does not actually analyse particle number fluctuations for generic models of the form (1), introduced in Ref. [8] as effective models for MBL systems. In particular, on p. 3 the Authors write “we will argue that the qualitative findings are generic”: do they mean with this that the conclusions can be extended to a more general class of effective models? Could they point out where is this argued in the manuscript?

8) The interpretation of the finite-size data on the number entropy presented by the Authors in this manuscript and in the previous publications [28-31] has been the subject of debate recently, in particular for what concerns the implications for the thermodynamics limit. In [Luitz et al, PRB 102, 100202(R), 2020] it is claimed that the double-logarithmic behaviour of the averaged number entropy is a transient effect, and that the distribution of the number entropy at infinite-time has tails that decay exponentially fast with L. Could the Authors comment on this point?

Requested changes

I would ask to the Authors to address the points 3,5,6,7 above by adjusting the manuscript.
I would ask them to comment on points 1,2,4,8 in the manuscript, if appropriate.

  • validity: good
  • significance: ok
  • originality: good
  • clarity: ok
  • formatting: good
  • grammar: excellent

Author:  Maximilian Kiefer-Emmanouilidis  on 2021-11-08  [id 1922]

(in reply to Report 1 on 2021-09-10)

First of all, we would like to thank the referee for the very careful and thorough evaluation of our work and the helpful comments. In the following, we will provide a detailed and point-by-point reply.

1) In the abstract and throughout the paper, the Authors insist on the fact that MBL systems are characterised by the presence of an exponential number of local charges. I believe that this statement is confusing (I interpret it as “exponentially-many in the system size L”). Integrability in MBL is usually understood in terms of linearly-many local conserved quantities, which are obtained as quasi-local rotations of the L conserved quantities associated to the quadratic, non-interacting part of the Hamiltonian. Could the Authors specify what they mean by “exponential number” here?

We follow here the discussion in the literature, see for example Huse et al. Phys. Rev. B 90, 174202 (2014) [Ref. 8 in our manuscript], where l-bit Hamiltonians as effective Hamiltonians for MBL phases were first introduced. In a fully localized many-body phase, all the eigenstates $\left| n \right\rangle$ ($2^N$ many in the case considered here) are localized. Therefore each projector $P_n = \left| n \right\rangle\left\langle n \right|$ is a local conserved charge: The system thus has exponentially many local charges.

We now have to think about fully characterizing these $2^N$ localized eigenstates by the $\eta$-operators (“l-bits”) in Eq. (1) of our manuscript instead of the projection operators $P_n$. To do so, let us just directly cite from the Huse et al. paper mentioned above (see page 4, left column): “Thus, we have argued that systems in the FMBL regime can be viewed as a type of “integrable” system, with Hamiltonian (1), which can be used to understand their dynamics. Traditional, translationally invariant integrable one-dimensional models of $N$ spins have $N$ conserved local densities. It appears that if you try to make other conserved quantities as composites (operator products) of these basic conserved densities, these are necessarily nonlocal operators of range $\sim N$. For a FMBL system, on the other hand, if we consider n l-bits near site $i$, out of products of these l-bits we can make $2n$ independent conserved quantities that are all localized near $i$. In this sense, fully many-body-localized systems have many more conservation laws that can affect local observables than do traditional translationally invariant integrable systems.”

Fully localized MBL (FMBL) phases can therefore - if such an l-bit description would in fact be correct - be viewed as described by ‘superintegrable’ models with exponentially many local charges instead of the linearly many in Bethe ansatz integrable models.

2) In Sec. 2 the Authors define the different Hamiltonians (i-iii) and write that they are used to determine the time evolution. I would clarify here how this time evolution is determined: is it via exact diagonalisation of the Hamiltonian and explicit calculation of the evolution operator? Given that the Authors consider quenches from a fixed pure initial state, could for example Krylov methods be used to have access to larger system sizes (L=18)?

We used exact diagonalization (ED) to determine the time-evolution of the system. We modified the text accordingly. Using Krylov methods is indeed a very powerful method to explore larger system sizes. However, it is computationally very demanding to reach time scales exceeding $\sim 10^4 1/J$ where $J$ is the hopping amplitude. Since we are investigating the dynamical properties of the system for large disorder, much longer time scales are needed already for system sizes $L>12$. We therefore selected ED as our main numerical method.

3) In Figs. 4 (d) and 5 (d) the Authors show the scaling with system size of the Hartley number entropy and of the number variance for model (iii), and claim that the curves are overlapping for the two larger system sizes. I suggest to modify (diminish) the y-range of these plots, to make this sister-size independence clearly evident from the plot. The same for Figs. 4 (c) and 5 (c).

We thank the reviewer for this comment. We added an inset to all these figures.

4) In Sec. 3.1 the Authors claim a different L dependence of the saturation values of the number entropy in models (i) and (iii): is it correct to say that these conclusions rely on the double-logarithmic fit of the number entropy behaviour in time? Is there any analytical argument justifying this scaling, similar to what one has for the log(t) behaviour of the entanglement entropy?

The different L dependence is directly evident from the numerical results for the saturation values of the considered entropies: the added insets show this even more prominently. A double-logarithmic fit is not required.

With regard to the second question: Yes, there is an analytical argument for the double-logarithmic scaling of the number entropy with time. In SciPost Phys. 8, 083 (2020) we have shown that $S_N \sim \mathrm{ln}(S)$ for non-interacting fermions. In Phys. Rev. Lett. 124, 243601 (2020) and Ann. Phys. (N.Y) 168481 (2021) we have then numerically demonstrated that this scaling - with some renormalization of the numerical factors - also holds in interacting cases and, in particular, for the disordered t-V model considered here.

5) The approximate model (iii) considered in this work is obtained neglecting the assisted and pair hopping-terms that appear in the Hamiltonian of model (i), when the latter is expressed in the Anderson basis. The numerical data presented in this work indicate that these terms play a relevant role when considering particle number fluctuations and therefore can not be neglected even when the interaction term is not too strong, at least in these finite sizes. Based on this, the Authors also claim: “as a consequence, it appears questionable if the microscopic model possessed an exponential number of exactly conserved local charges”. This implication is not clear to me. At the level of the local conserved quantities, the approximation scheme discussed in this work corresponds to projecting the conserved operators onto a small subspace, that is the kernel of the commutator . This is a quite substantial approximation to the “true” conserved quantities: its failure in reproducing certain features of the dynamics is in my view not sufficient to question overall the existence of local (non-approximated) conserved operators, and thus to question MBL integrability. Could the Authors make their argument more stringent?

Any model can be brought into the form of the Hamiltonian (1), for example, by successive Schrieffer-Wolff transformations. What is supposed to be special in the MBL case is that all the $\eta$-operators are local and that the interactions between them are exponentially decaying.

In Jour. Stat. Phys. 163:998-1048 (2016) it is claimed that such a successive diagonalization to an effective Hamiltonian with local conserved charges is fully under control. However, no explicit method to do so numerically is known and the results in the paper above are under debate in the community, see for example the recent symposium "MBL Dead or Alive?" (https://itsatcuny.org/calendar/mbl21).

We do not claim to resolve this issue here; our goal is much more modest. As we state explicitly in the title, we investigate a special, simple effective model for which we do know for sure that the $\eta$-operators are local and which is based on the original idea that the $\eta$-operators for small interaction strengths are simply the dressed Anderson orbitals. We also note that exactly this model has been claimed to effectively solve the MBL dynamics at strong disorder, see for example De Tomasi et al, Phys. Rev. B99, 241114 (2019) [Ref. 34 in our updated manuscript]. We show here that this is not the case, at least not for long times.

As far as the generality of the conclusions based on this model is concerned: It is clear that in any model of the type (1) with exponentially-many local conserved charges, the number fluctuations have to converge to a finite value in the thermodynamic limit. This qualitative picture does not change by a further renormalization of the Anderson orbitals. If the number fluctuations do not converge to a finite value, then such a description is not possible and MBL does not exist.

While it can never be excluded based on numerical results that even at very strong disorder and small interactions an anomalously large renormalization of the Anderson orbitals takes place which makes them immediately much more extended, this is unexpected and would need to be explained if MBL is ‘not dead’. At the very least, such an extremely strong renormalization would add to the mystery of the very large critical disorder strength for the transition into the MBL phase which latest studied now put at $D/J > 80$.

6) In the Introduction, it is written “if one wants to take into account the normalisation of the orbitals [….], then one has to ensure that this renormalisation does not ultimately lead to delocalised orbitals”. Similarly, in the Conclusion the Authors write “The question then is, whether […] such a normalization ultimately leads to charges becoming non-local”. In fact, this question has been the subject of several previous works, cited in the introduction of the manuscript. In particular, Ref. [11] claimed a proof of the fact that the renormalisation procedure does not ultimately lead to delocalized orbitals, at least for a particular class of locally-interacting 1d spin models. The same is claimed based on approximate perturbative arguments for fermonic Hamiltonians of the type discussed in this manuscript. I believe that this previous work should be accounted for when raising this point, even more so if the Authors have reasons to challenge those results or interesting intuitions on possible loopholes in the proof or in the perturbative arguments: at the moment, the way the discussion is formulated gives the impression that this issue has remained completely unadressed in the literature.

See answer of point 5. We have added a reference to the Jour. Stat. Phys. article by Imbrie.

7) I would also suggest to be more explicit one the fact that this manuscript is dealing with a specific approximation, the one leading to model (iii), and does not actually analyse particle number fluctuations for generic models of the form (1), introduced in Ref. [8] as effective models for MBL systems. In particular, on p. 3 the Authors write “we will argue that the qualitative findings are generic”: do they mean with this that the conclusions can be extended to a more general class of effective models? Could they point out where is this argued in the manuscript?

We believe that this point is cleary addressed in the title of our manuscript. See also point 5.

8) The interpretation of the finite-size data on the number entropy presented by the Authors in this manuscript and in the previous publications [28-31] has been the subject of debate recently, in particular for what concerns the implications for the thermodynamics limit. In [Luitz et al, PRB 102, 100202(R), 2020] it is claimed that the double-logarithmic behaviour of the averaged number entropy is a transient effect, and that the distribution of the number entropy at infinite-time has tails that decay exponentially fast with L. Could the Authors comment on this point?

We have covered the concerns of Luitz and Bar Lev in detail in our publication Phys. Rev. B 103, 024203 (2021). Furthermore, in a preprint by Morningstar, Colmenarez, Khemani, Luitz, Huse arXiv:2107.05642, the authors themselves have weakened their previously made arguments “We believe that, by doing so, we are beginning to try to remedy a methodological error that has been made by much of the MBL research community, the majority of the present authors included.”. We also note that in the same paper the critical disorder strength is revised from $D \sim 16$ to $D>28$ and in arXiv:2108.10796, using the same method as in arXiv:2107.05642 but for larger systems, to $D>80$. We became aware of these manuscripts after our submission and have updated the reference list accordingly. See also the answer to point iii) of the second reviewer.

---

## Round 1 · Referee Report · Anonymous (Referee 1) · 2021-9-28

Strengths

1) Thorough numerical analysis on three related models.
2) New questions are posed about local conserved charges in many-body localisation.

Weaknesses

1) Some points need more detailed description.

Report

In this paper one-dimensional quantum models are considered in the presence of disorder and their dynamical properties are studied through numerical exact diagonalization. In particular the entanglement entropy, the number entropy and the time-averaged particle number fluctuations are investigated and compared for the following three models.

(a) The t-V model, with fermionic hopping, nearest-site interaction (V) and a random potential term (D)

(b) An effective model, which is obtained from (a) by keeping interaction terms which are diagonal in the V=0 basis. This model is expected to be accurate in the small-V regime.

(c) The non-interacting (V=0) version of the model (a), which describes Anderson localisation.

According to the numerical results the entanglement entropy for (a) and (b) for small V has the same type of logarithmic increase in time, while (c) has a saturated value. On contrary, the time-dependence of the Hartley entropy and the number entropy is different for the three models. Similar conclusion is obtained for the particle number fluctuations, which increases for (a), but saturates for the effective model (b). From this the authors conclude that during the transformation with the Anderson basis the effect of the off-diagonal terms can not be neglected. These either renormalise the orbitals in the Anderson model with local conserved charges, or these charges became non-local.

This paper represents the continuation of a series of papers of the authors about many-body localization [28-31]. The subject of the present paper is interesting, the obtained results could give new impulses to clarify this phenomena more deeply. The paper is basically well written. I suggest publication of this work after the authors have successfully addressed the following points.

i) The authors should comment on the numerical methods they used. In Fig.11 I expect free-fermionic methods were used.

ii) A comment on the necessary number of disorder realizations would be in order.

iii) For the number entropy and the particle number fluctuations for model (a) double-logarithmic time-dependence fit is used. Such type of scaling works at infinite disorder fixed points (Phys. Rev. B85, 094417 (2012), Phys. Rev. B 93, 205146 (2016)). Can the authors explain such type of scaling in this problem?

iv) Somewhat related point: in Phys. Rev. B102, 100202(R) (2020) the double-logarithmic increase of the number entropy in the thermodynamic limit is debated. The authors should comment on this.

v) Small point: a few typos should be fixed. see an an, ...

  • validity: high
  • significance: high
  • originality: high
  • clarity: high
  • formatting: excellent
  • grammar: excellent

Author:  Maximilian Kiefer-Emmanouilidis  on 2021-11-08  [id 1921]

(in reply to Report 2 on 2021-09-28)
Category:
answer to question

First of all, we would like to thank the referee for the very careful and thorough evaluation of our work and the helpful comments. In the following, we will provide a detailed and point-by-point reply.

i) The authors should comment on the numerical methods they used. In Fig.11 I expect free-fermionic methods were used.

We thank the reviewer for pointing this out. In general, we have used exact diagonalization (ED) to determine the time-evolution of the system. For Fig.11 we used free-fermionic methods; we modified the text accordingly.

ii) A comment on the necessary number of disorder realizations would be in order.

We checked that the amount of disorder realizations is sufficient by calculating data for different numbers of realizations. See the attached file and point 5) in the list of changes.

iii) For the number entropy and the particle number fluctuations for model (a) double-logarithmic time-dependence fit is used. Such type of scaling works at infinite disorder fixed points (Phys. Rev. B85, 094417 (2012), Phys. Rev. B 93, 205146 (2016)). Can the authors explain such type of scaling in this problem?

We studied the number entanglement properties of the tight-binding model with disordered hopping amplitudes (off-diagonal disorder) as considered in Phys. Rev. B 93, 205146 (2016) in detail in SciPost Phys. 8, 083 (2020). In the latter paper we derive analytical bounds which show that a scaling of the entanglement entropy $S\sim \mathrm{lnln}t $ implies an $S_N \sim \mathrm{lnlnln} t$ scaling of the number entropy. Furthermore, we considered in Phys. Rev. B 103, 024203 (2021) the argument by Luitz and Bar Lev in Phys. Rev. B102, 100202(R) (2020) that the exponential tail of the number entropy distribution implies that $S_N$ saturates by comparing to the number entropy distribution in the off-diagonal case. Here we found that the tail in the distribution for the off-diagonal disorder case – a model which we know for sure is not localized - decays even faster than in the ‘MBL case’.

iv) Somewhat related point: in Phys. Rev. B102, 100202(R) (2020) the double-logarithmic increase of the number entropy in the thermodynamic limit is debated. The authors should comment on this.

We have extensively discussed the points made by Luitz and Bar Lev in our publication Phys. Rev. B 103, 024203 (2021), see also point (iii) above. In addition, we have learned that when averaging over all initial product states a 1/L correction is always present which counteracts the increase of the saturation value of the number entropy with L. This effect can already be seen in the Anderson model and is discussed in the appendix of the manuscript under review. These effects were not taken into account in the analysis presented in Phys. Rev. B 103, 024203 (2021).

v) Small point: a few typos should be fixed. see an an, ...

We thank the reviewer for pointing these out and have corrected them.

Attachment:

L12_D20_V2p0_comparison_sample.pdf

---

## Editorial Decision

resubmitted